# Cognitive Assessment Tools Recommended in Geriatric Oncology Guidelines: A Rapid Review

Gina Tuch [1,*], Wee Kheng Soo [2,3,4], Ki-Yung Luo [5], Kinglsey Frearson [5], Ek Leone Oh [5], Jane L. Phillips [6], Meera Agar [7] and Heather Lane [5]

1  Department of Geriatric Medicine, Alfred Health, Melbourne, VIC 3004, Australia
2  Eastern Health Clinical School, Monash University, Box Hill, VIC 3128, Australia; kheng.soo@outlook.com
3  Cancer Services, Eastern Health, Box Hill, VIC 3128, Australia
4  Aged Medicine Program, Eastern Health, Box Hill, VIC 3128, Australia
5  Sir Charles Gairdner Hospital, Nedlands, WA 6009, Australia; Ki-Yung.Luo@health.wa.gov.au (K.-Y.L.); Kingsley.Frearson@health.wa.gov.au (K.F.); Leone.Oh@health.wa.gov.au (E.L.O.); Heather.Lane@health.wa.gov.au (H.L.)
6  Queensland University of Technology, Brisbane City, QLD 4000, Australia; jane.phillips@qut.edu.au
7  University of Technology Sydney, Ultimo, NSW 2007, Australia; meera.agar@uts.edu.au
*  Correspondence: g.tuch@alfred.org.au or tuchg01@gmail.com

**Abstract:** Cognitive assessment is a cornerstone of geriatric care. Cognitive impairment has the potential to significantly impact multiple phases of a person's cancer care experience. Accurately identifying this vulnerability is a challenge for many cancer care clinicians, thus the use of validated cognitive assessment tools are recommended. As international cancer guidelines for older adults recommend Geriatric Assessment (GA) which includes an evaluation of cognition, clinicians need to be familiar with the overall interpretation of the commonly used cognitive assessment tools. This rapid review investigated the cognitive assessment tools that were most frequently recommended by Geriatric Oncology guidelines: Blessed Orientation-Memory-Concentration test (BOMC), Clock Drawing Test (CDT), Mini-Cog, Mini-Mental State Examination (MMSE), Montreal Cognitive Assessment (MoCA), and Short Portable Mental Status Questionnaire (SPMSQ). A detailed appraisal of the strengths and limitations of each tool was conducted, with a focus on practical aspects of implementing cognitive assessment tools into real-world clinical settings. Finally, recommendations on choosing an assessment tool and the additional considerations beyond screening are discussed.

**Keywords:** aged; medical oncology/standards; geriatric assessment/methods; cognition/physiology; cognitive dysfunction/diagnosis; dementia/diagnosis; screening; clinical decision-making

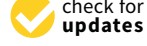



## 1. Introduction

Cognitive impairment affects various aspects of care in older adults with cancer, such as the ability to participate in clinical decision-making, cope with treatment, and self-manage treatment regimens, as well as being associated with worse cancer and non-cancer outcomes [1–5]. Cancer and cognitive impairment are both associated with ageing [6]. Three out of every five new cancer diagnoses are in people aged 65 years or older, and 15–48% of patients 70 years and older with cancer have impairment detected on cognitive testing [6]. However, cognitive impairment is often subtle and easily overlooked by clinicians on a routine clinical assessment [7]. Therefore, a systematic approach to cognitive assessment in older adults with cancer is required [7].

At diagnosis, a patient is expected to participate in often complex decision-making that requires the ability to consider concepts which are multidimensional, as well as be able to weigh up potential trade-offs, such as quality of life and anticipated prognosis [1,2,5]. Moreover, patients need to consider decisions which have significant implications for future care, such as naming a power of attorney or substitute decision-maker, completing a will,

and participating in advance care planning conversations on future care decisions [3,5,6,8]. During the treatment phase, cognitive limitations may increase the risk of medication administration errors, amplified in the context of self-management of oral chemotherapy regimens at home, as well as making the logistics of attending regular appointments more challenging [4]. Adapting to changes in daily routine due to treatment-associated side effects or reduction in functional ability may exceed the mental flexibility of those with executive impairment. This can have profound impacts such as increased hospitalizations during treatment, reduced ability to report complications, increased need for caregiver support potentially contributing to caregiver distress, and even compromised independence with potential increased relocation to supported accommodation [6,8]. It is not unexpected that cognitive impairment is associated with worse cancer and non-cancer outcomes including increased toxicity from the treatment, lower completion rate of treatment, worse overall survival, increased caregiver burden, cognitive decline with the treatment, and poorer patient-reported outcomes [1–4].

The term cognitive impairment encompasses a variety of neurocognitive conditions, including dementia, delirium, and Cancer-Related Cognitive Decline (CRCD). Cognitive impairment disorders range from Mild Cognitive Impairment (MCI) to dementia. MCI is a modest cognitive decline in one or more cognitive domains in history and formal testing, without change in functional independence (in the absence of other mental disorders) [9]. Dementia is a more substantial cognitive decline in one or more cognitive domains in history and formal testing, with an associated reduction in functional independence (in the absence of other mental disorders) [9]. Delirium is an acute confusional state characterized by deficits in attention and concentration, which tends to fluctuate during the course of the day and is associated with physiological stressors [10]. A pre-existing cognitive impairment increases the risk of delirium with physiological stressors such as chemotherapy and surgery [1,8]. Finally, CRCD refers to changes in cognition that occur during or after a cancer treatment and tend to impact domains of executive function, processing speed, memory, and attention [11,12]. This review focuses on assessing for pre-existing cognitive impairment disorders (e.g., MCI or dementia) in older adults with cancer, as distinct from delirium or CRCD [6].

Cognitive assessment is typically incorporated into the broader Geriatric Assessment (GA) of the older person with cancer. The GA evaluates various domains (cognition, comorbidities, function, mobility, nutrition, polypharmacy, psychological, and social support) with a view to guide interventions to improve the care of and outcomes for older adults with cancer [13,14]. GA has been demonstrated to alter the initial oncology treatment plan in at least a quarter of patients (median 28%; range 8–54%), with non-oncological interventions being recommended in almost three-quarters of patients (median 72%; range 26–100%) [15]. Recent randomized controlled trials demonstrated improvements in quality of life, unplanned hospital admissions, and reduced chemotherapy toxicity and early treatment discontinuation, confirming the importance of GA in older cancer patients [16–19]. In addition, several Oncological Societies internationally have published guidelines on GA, which include recommendations for cognitive assessment or screening [1,20–22]. If cancer care services do not screen for pre-existing cognitive impairment, this limits the information available to truly inform decision-making for older people and their families, as well as clinicians, and may lead to inadequate support for the upcoming cancer journey [2,3]. At the same time, many older people may not have heard of cognitive screening, nor seen themselves as at risk of impairments in thinking or memory, therefore assessing cognition needs to be approached with great sensitivity.

This rapid review seeks to evaluate cognitive assessment tools routinely used in older adults with cancer to provide clinical and practical information for cancer care clinicians. While the increased assessment of cognition in oncology is advocated, all cognitive assessment tools have inherent strengths and limitations. Awareness of these characteristics will aid the selection of a tool appropriate to the clinical setting and reduce the chances of misinterpretation of assessment results.

## 2. Materials and Methods

This narrative literature review consists of two components: (1) Identification of the recommended cognitive assessment tools; and (2) focused review of the evidence underpinning the identified tools.

### 2.1. Part 1: Identification of Cognitive Assessment Tools

A search was conducted to identify recent Oncology guidelines focused on geriatric assessment, both within the published literature and the web-based "grey literature". Searches were limited to guidelines published in the last 5 years.

#### 2.1.1. Published Resources

PubMed and Ovid Medline were searched using the terms in Table 1.

**Table 1.** Search strategy published resources.

| | |
|---|---|
| | Cancer*[ti/ab] OR Neoplasm*[ti/ab] OR Oncolog*[ti/ab] |
| AND | geriatric*[ti/ab] OR elderly*[ti/ab] OR "older adult*"[ti/ab] OR "senior adult*"[ti/ab] OR "older patient*"[ti/ab] |
| AND | assessment*[ti/ab] OR "evaluation*[ti/ab] |
| AND | guideline*[ti/ab] OR "position statement"[ti/ab] OR "consensus statement" [ti/ab] or recommendation*"[ti/ab] |

Ti/ab = title or abstract; *, demonstrate the truncated search term.

#### 2.1.2. Web-Based Resources

To identify guidelines that may not be published in the peer-reviewed literature, a systematic search using predefined search terms was undertaken using the Google search engine to identify the most commonly viewed resources. The search strategy was refined after piloting to (cancer OR oncology) AND (geriatric OR elderly OR "older adult" OR "senior adult") AND (guideline OR "position statement" OR "consensus statement" or recommendation). The first 100 websites featured in the results for the search were screened.

#### 2.1.3. Inclusion Criteria

- Published between 1 July 2016 and 1 July 2021;
- Available in English;
- Guidelines, position statements, consensus statements or recommendations;
- Primary focus is the assessment of older adults with cancer;
- Authored or published by a national or international Medical or Oncological Society or Organization; and
- Refer to specific cognitive assessment tools.

#### 2.1.4. Exclusion Criteria

- Opinion pieces, research articles, and review articles;
- Authored or published by a single institution;
- Related to a single cancer type or groups of cancers of a single body system or organ; and
- Tools designed specifically for screening or assessing acute confusional states or delirium.

Where there was uncertainty about whether an article/website met the inclusion/exclusion criteria, this was discussed by the research team, and a consensus decision was reached.

Any tools or instruments recommended for cognitive assessment or cognitive screening in these guidelines were identified. For this rapid review, the following operational definitions were adopted:

- Clinical practice guidelines: Evidence-based statements that include recommendations intended to optimize patient care and assist health care practitioners to make decisions on the appropriate health care for specific clinical circumstances [23].

- Cognitive assessment tool: Any instrument, tool or survey, developed or utilized to assess or screen cognitive function in adults.

### 2.2. Part 2: Focused Review of the Evidence Underpinning the Identified Tools

Characteristics of interest for the cognitive assessment tools were determined by an iterative consensus process among the authors. A focused search using PubMed and Medline was conducted for each cognitive assessment tool identified in Part 1 to establish their characteristics (Table 2). This was supplemented by the Google search engine if the relevant information about the cognitive assessment tool was not available in the peer-reviewed literature. Finally, a narrative synthesis was used to describe the findings and generate practice points to support clinicians in determining the best tools to use in clinical practice.

**Table 2.** Characteristics of cognitive assessment tools.

| Psychometric Considerations for Clinical Use |
| --- |
| Validated populations |
| Validation in older adults with cancer |
| Reliability |
| Sensitivity and specificity for detecting cognitive impairment |
| Effect of literacy or education |
| Effect of visual impairment |
| **Practical Considerations** |
| Cognitive domains assessed |
| Completion time |
| Training recommendations and frequency |
| Cost |
| Accessibility |
| Available in languages other than English |
| Available in alternate versions for repeat testing |
| Suitability for telehealth consultations |

### 3. Results

Eight guidelines on the assessment of older adults with cancer (Table 3) were identified from the search (Figure 1), from which six cognitive assessment tools were identified (Tables 4 and 5). The guidelines recommended between one and five cognitive assessment tools. The Spanish Society for Medical Oncology (SEOM) 2018 [24] and National Comprehensive Cancer Network (NCCN) guidelines [22,25] endorsed a single brief cognitive screening tool, with recommendations for further cognitive assessment if screening was abnormal [22,24,25]. The International Geriatric Oncology Society (SIOG) COVID-19 guidelines recommended the Blessed Information Memory Concentration (BOMC) in the context of telehealth consultations [26]. Other guidelines suggested several cognitive assessment tools. The Mini-Mental Status Examination (MMSE) and Montreal Cognitive Assessment (MoCA) were most commonly recommended, closely followed by the Mini-Cog. Consistency was seen across most of the guidelines, with only the SEOM recommending a tool that was not recommended by any other guideline, the Short Portable Mental Status Questionnaire (SPMSQ) [24]. Only the American Society of Clinical Oncology (ASCO) [1] and Young International Society for Geriatric Oncology (SIOG) [21] guidelines commented specifically on the cut-off test scores indicating concern, and these are discussed in the relevant sections. The commonly utilized cognitive assessment tools encompassed variable combinations of cognitive domains, such as orientation (temporal and spatial), memory registration

and recall, attention and concentration, language (written and verbal), visuo-spatial and visuo-constructional, and executive function.

**Table 3.** Guidelines on the assessment of older adults with cancer.

| Guidelines | Organization | Year of Publication | Recommended Tools |
|---|---|---|---|
| Geriatric assessment in daily oncology practice for nurses and allied health care professionals: Opinion paper of the Nursing and Allied Health Interest Group of the International Society of Geriatric Oncology (SIOG) [20] | Nursing and Allied Health Interest Group of SIOG (International Society for Geriatric Oncology) | 2016 | Mini-Cog, MMSE, MoCA |
| NCCN Clinical Practice Guidelines in Oncology: Older adult oncology, version 2.2017 [25] | NCCN (National Comprehensive Cancer Network) | 2017 | Mini-Cog, MMSE, MoCA |
| Practical assessment and management of vulnerabilities in older patients receiving chemotherapy: ASCO guideline for geriatric oncology [1] | ASCO (American Society of Clinical Oncology) | 2018 | Mini-Cog, BOMC, MMSE, MoCA |
| General recommendations paper on the management of older patients with cancer: The SEOM geriatric oncology task forces position statement [24] | SEOM (Spanish Society for Medical Oncology) | 2018 | SPMSQ |
| What every oncologist should know about geriatric assessment for older patients with cancer: Young International Society of Geriatric Oncology position paper [21] | Young SIOG (Young International Society for Geriatric Oncology) | 2018 | Mini-Cog, MMSE, MoCA, BOMC, Clock-drawing |
| NCCN Clinical Practice Guidelines in Oncology: Older adult oncology, version 1.2019 [22] | NCCN (National Comprehensive Cancer Network) | 2019 | Mini-Cog, MMSE, MoCA |
| Adapting care for older cancer patients during the COVID-19 pandemic: Recommendations for the International Society for Geriatric Oncology COVID-19 Working Group [26] | SIOG (International Society for Geriatric Oncology) | 2020 | BOMC |
| Comprehensive geriatric assessment in older adults with cancer: Recommendations by the Italian Society of Geriatrics and Gerontology [27] | SIGG (Italian Society of Geriatrics and Gerontology) | 2021 | MoCA, MMSE |

**Table 4.** Psychometric considerations for clinical use.

| Tool Name | Year Published | Validated Population | Validated in Older Adults with Cancer? | Reliability | Sensitivity and Specificity at Standard Cut-Points | Effects of Literacy or Education | Effects of Visual Impairment |
|---|---|---|---|---|---|---|---|
| BOMC [28] | 1983 | Validated in a variety of clinical settings [29,30] | No | Test-retest reliability correlation coefficient: 0.77 [31] | Sensitivity 78.5–83%, specificity 77–100% to detect dementia [32] | Not considered sensitive to educational level [33] | No visual elements |
| CDT [34] | 1963 | Widely validated in multiple countries and various neurological conditions [35–38] | Comparable with MMSE results in cancer population [39] | High inter-rater and test-retest reliability [40] | Mean sensitivity 85% and specificity 85% to detect dementia [40] | Not valid in those with ≤4 years of formal schooling [41] | Requires adequate vision |
| Mini-Cog [42] | 2000 | Validated in a variety of community and clinical settings and populations [43] | No | High inter-tester reliability [44] | Sensitivity 76% and specificity 73% to detect dementia [45] | Not recommended in those with ≤5 years of formal education [46] | Requires adequate vision |
| MMSE [47] | 1975 | Extensively validated in many countries, populations, and in a range of different neurological and neurocognitive conditions [48–52] | Validated in cancer populations [53] | Variability reported in the test-retest reliability [47,54,55] | Sensitivity 0.85, specificity 0.90 for detecting dementia [56] | Scores decrease with advancing age and less education | Includes writing, drawing, and reading tasks |
| MoCA [57] | 2005 | Extensively validated in many countries, populations, and in a range of neurological and neurocognitive conditions [58–62] | No | High test-retest reliability [63] | MCI sensitivity 90%, specificity 87% *AD sensitivity 100%, specificity 87% [57] | Add 1 point if ≤12 years of education [57] | Hearing and visual impairment affect the MoCA performance [64] |
| SPMSQ [65] | 1975 | Assessed in hospital inpatients, nursing homes, Finish, Singaporean, Iranian populations [66] | No | Test-retest reliability: 0.8–0.83 Interrater reliability: 0.62–0.87 [67] | Sensitivity 55–85.7% and specificity 78.9–96% for detecting dementia [32] | Subtract 1 from error score if grade school education, add 1 to error score if education beyond high school [65] | No visual elements |

* AD = Alzheimer's dementia.

**Table 5.** Practical considerations for clinical use.

| Tool Name | Cognitive Domains | | | | | | | Score Interpretation | Completion Time | Training Requirements | Fee | Copyright, Open Access, and Permission to Reuse | Languages | Alternate Versions | Telehealth Version |
|---|---|---|---|---|---|---|---|---|---|---|---|---|---|---|---|
| | Memory | Visuospatial | Orientation | Attention | Language | Praxis | Executive | | | | | | | | |
| BOMC [28] | ✓ | | ✓ | ✓ | | | | ≤7 = Normal ≥8 = Abnormal | 2–3 min [68] | No training recommended | Free usage for healthcare professionals | Copyrighted | Three languages: English, Portuguese, and Spanish [69,70] | No | Suitable for telephone consultation |
| CDT [34] | | ✓ | | | | | ✓ | Multiple different scoring methods [36,40] | 1–5 min [71] | Can be administered by non-trained professionals [72] | Freely available | Open access | Does not require translation | No | Suitable for video consultation |
| Mini-Cog [42] | ✓ | ✓ | | | | | ✓ | ≥3 = Lower likelihood of dementia [73] | 3 min [69] | Training not required, 90% concordance between "expert" and regular raters [44] | No cost | Copyrighted, but may be used without permission in clinical and educational settings [73] | Six languages: English, Spanish, Portuguese, Chinese, Malay, and Arabic [73] | No | Suitable for video consultation |
| MMSE [47] | ✓ | ✓ | ✓ | ✓ | ✓ | ✓ | | ≥24 = Normal [47] | 10–15 min [47] | No specific training is recommended | There is a cost to purchasing tests [74] | Copyrighted | 75 languages [74] | Many variations are published [75,76] | Validated telephone versions are available [77] |
| MoCA [57] | ✓ | ✓ | ✓ | ✓ | ✓ | | ✓ | >26 = normal ≤26 = Abnormal [57] | 10 min [54] | Mandatory training and certification program. Retraining recommended twice yearly [78] | USD 150 for training. No cost for using the test for clinical or teaching purposes [78] | Copyrighted, available for use without permission for clinical and teaching purposes [78] | Paper version: Nearly 100 languages App version: Five languages [78] | Three versions are available for repeat testing, if retesting within 3 months [78] | Video consultation with modified instructions. Abbreviated telephone version available [78] |
| SPMSQ [65] | ✓ | | ✓ | ✓ | | | | 0–3 errors = normal [79] | 5–10 min [80] | Can be applied without formal training [81] | Freely available | Open access | Three languages: English, Spanish, and Iranian [66,82] | No | Suitable for telephone consultation |

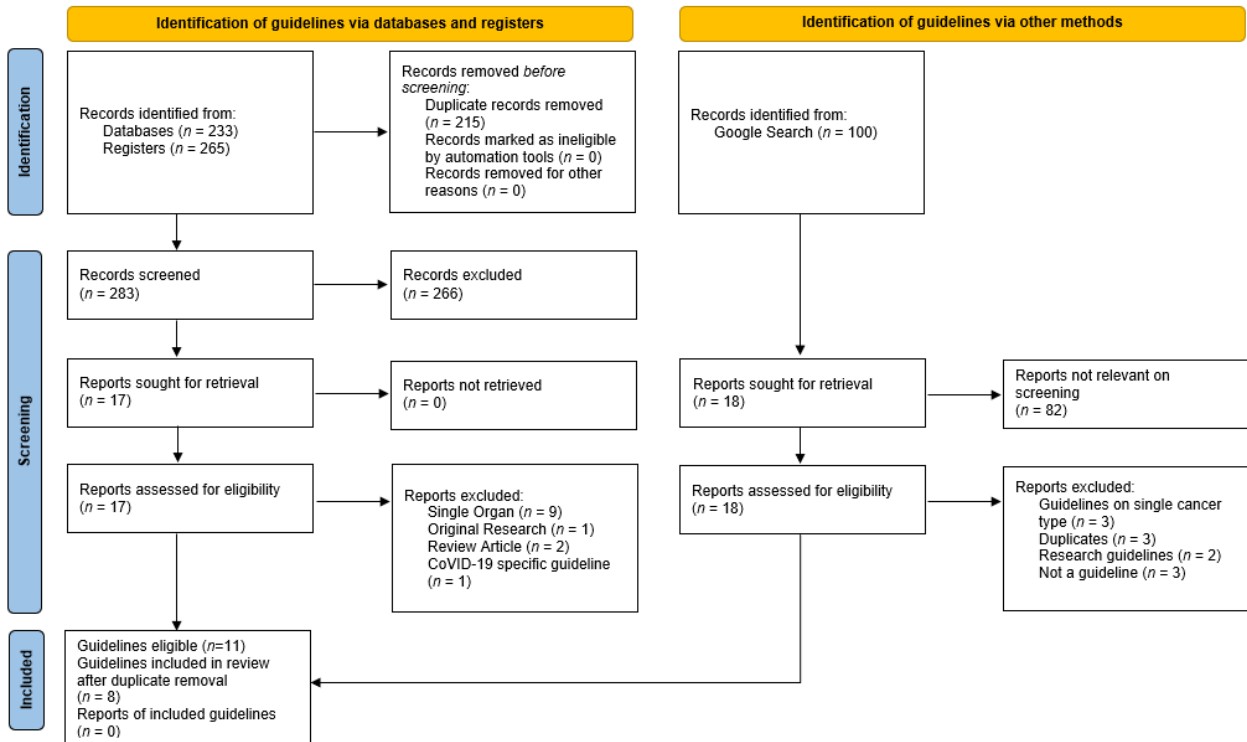

**Figure 1.** Flow chart database search demonstrating the records identified, screened, and included with reasons for exclusion identified.

### 3.1. Blessed Information Memory Concentration

The Blessed Information Memory Concentration (BOMC) [28], also known as the six-item Cognitive Impairment Test (6-CIT) and the short Orientation-Memory-Concentration test, is a six-item cognitive assessment tool that can be completed in 2–3 min [68]. It is limited to assessing the three domains described in its title [28], thus not testing executive function. Typically, a score of 8 or more is considered abnormal [28]. However, the ASCO and Young SIOG guidelines recommend using lower cut-off scores of 6 and 4, respectively [1,21], which will increase the detection of milder cognitive problems. It has not been translated or validated as extensively as other tools. Although copyrighted, it is freely available to healthcare professionals and does not require specific training to be used. It has no visual elements, thus no adaptation is needed for telephone consultations or for those with visual impairment.

### 3.2. Clock Draw Test

The Clock Draw Test (CDT) [34] is a well-established and widely validated cognitive assessment tool. It is very brief, freely available, and does not require translation into other languages. It tests visuo-spatial skills and executive function, but not memory [83]. It is considered a cognitive screening tool for identifying those who should undergo further cognitive assessment rather than a diagnostic tool [84,85]. A variety of scoring methods are used, complicating the interpretation of the literature [36]. Adaptations using a digital clock have been studied [83,86].

### 3.3. Mini-Cog

The Mini-Cog incorporates the CDT and adds the three-item recall, thus adding a memory domain [42]. Moreover, it has a standardized scoring system. Although, a score of less than 3 is typically used to define an abnormal result [42], the Young SIOG guidelines

recommend using a score of less than 4, to improve the detection of milder degrees of cognitive impairment [21]. It was developed as a screening tool for cognitive impairment in multi-lingual communities [42] and is available in several languages [73]. Minimal language interpretation is required, hence it can be easily used with an interpreter in additional languages [42]. It has been widely validated in different countries and populations [43,87–89]. Furthermore, it is copyrighted, but is freely available for use, reproduction, and distribution in clinical and educational settings without permission, and no formal training is required [73]. Its brevity and ease of use make it popular in the primary care and non-specialist secondary care setting.

### 3.4. Mini-Mental Status Examination

The Mini-Mental Status Examination (MMSE) is a well-established test [47]. It is widely studied with validation in multiple countries, languages, and clinical settings [48–52]. Many variations are available and have their own validation data. It is one of the longer tests and covers a range of cognitive domains, but not executive function [47]. Moreover, outcomes vary with education [90] and deficits may not be identified in those with high educational backgrounds [91]. It is copyrighted, thus cost and accessibility may limit its organizational use.

### 3.5. Montreal Cognitive Assessment

The Montreal Cognitive Assessment (MoCA) is one of the longer assessment tools in the guidelines, taking approximately 10 minutes to complete [57]. It covers a wide range of cognitive domains, including executive function [57]. It has been extensively validated in multiple languages and populations [58–61], as well as in varied neurological and neurocognitive disorders [92–95]. It is available in close to 100 languages, with adapted forms suitable for telephone administration (T-MoCA) [78]. It has high sensitivity and reasonable specificity in a population with dementia of Alzheimer's type [57]. Moreover, unlike the other tools, it has good sensitivity for detecting mild cognitive impairment [57]. There is a training fee, and training requirement which, once completed, all versions of the tests become available for use, reproduction, and distribution without permission for clinical and teaching purposes [57].

### 3.6. Short Portable Mental Status Questionnaire

The Short Portable Mental Status Questionnaire (SPMSQ) [65] is a brief 10-item cognitive screening tool. It is less studied than the other identified tools. However, it is validated in its Spanish version [82] and in a number of populations [66,79]. Similar to the other brief tools, fewer cognitive domains are covered (memory, attention, and orientation) [65]. It is freely available, can be applied without formal training, and can be used via telephone while retaining comparable sensitivity for the detection of dementia [96].

## 4. Discussion

It can be challenging for cancer care clinicians to choose from the plethora of the available cognitive assessment tools. Of the six cognitive assessment tools that were recommended by geriatric oncology guidelines, the MOCA [57], MMSE [47], and Mini-Cog [42] were the most frequently endorsed tools. However, there is no ideal cognitive assessment tool, each has advantages and limitations. It is important for clinicians to be aware of the strengths and limitations of each tool and the neurocognitive domains they cover. This rapid review found that executive function is a notable gap in both the MMSE [47] and BOMC [28]. Moreover, the BOMC lacks visuo-constructional/visuo-spatial assessment. We hope this review will aid clinicians in considering which tool or tools might best meet their needs, taking into account the demographics of their patient population, access to other services, and the practicalities of their clinical setting. While we only identified validation studies of the CDT [34] and MMSE [47] in cancer populations [39,53], other tools have been validated in large populations that will include older people with cancer.

An initial consideration is whether a brief cognitive screen or a more detailed cognitive assessment is required and practical. The identified tools include brief screening tools (e.g., Mini-Cog or CDT) and short tools (e.g., BOMC or SPMSQ) that do not require specific training. In a busy oncology clinic, these may be most feasible. If a more detailed cognitive assessment is accessible through referral pathways, then a brief screen is likely to be adequate. However, in geriatric assessment clinics, clinicians may well prefer to utilize one of the longer recommended tools, the MoCA or MMSE, to evaluate a broader range of cognitive domains.

The choice of cognitive assessment tool should be influenced by the demographics of the patient population, as patient factors must be considered when undertaking any cognitive assessment. Patients from culturally and linguistically diverse backgrounds require tools validated in these populations and languages. The MoCA, MMSE, and Mini-Cog have been translated and validated in many languages [73,74,78]. The Mini-Cog has shown to be useful and feasible for cognitive screening across ethnically diverse groups and those with lower literacy and education levels [97,98]. Educational level does affect the cognitive assessment performance across all of these tools, except the BOMC, which is reportedly not sensitive to educational level [33]. For those with a higher educational background, a more sensitive tool such as the MoCA may be required to demonstrate the underlying cognitive impairment. Moreover, the MoCA has greater sensitivity and specificity for the detection of MCI in comparison to other tools [2,57,99–102]. Furthermore, visual and hearing deficits need to be factored into the tool choice. The BOMC has no visual elements, while the CDT is primarily visual. Therefore, it may be suitable when substantial hearing difficulties are present.

Pragmatic considerations will often determine the cognitive assessment tool chosen in a particular setting. An organization may have a preferred tool or may not have licensing for certain tools. In addition, clinicians may not have training for particular tools. For telephone consultations, the BOMC or SPMSQ can be used without adaptation, while the Mini-Cog or CDT cannot be utilized. If a tool has been utilized in an individual before, repeating the same tool allows for a longitudinal comparison. However, alternate versions should be used if the tool is repeated at time intervals below the recommended test-retest interval to counteract the practice effect [103,104]. There may be a geographical variation in the cognitive assessment tool preference. Observationally, the BOMC seems to be more widely utilized in North America than elsewhere, and is recommended in the ASCO guidelines [1], while the SPMQ is only endorsed by the SEOM [24], which may relate to local preferences for this tool that are available and validated in Spanish [82].

A brief screening tool such as the Mini-Cog could be implemented into routine oncology clinic appointments for older people. Abnormal screening results could be followed by a more detailed assessment or on-referral. Based on our own practices, the authors recommend the Mini-Cog for brief screening and the MMSE or MOCA for more detailed testing where time permits. Moreover, this selection corresponds with the most endorsed tools by the guidelines. Where screening via phone is required, the BOMC could be used rather than the Mini-Cog and the telephone version of the MoCA could be used if more detailed testing is required. Clinicians who are assessing cognition in older adults frequently may wish to familiarize themselves with a small number of tools to provide a cognitive screening "tool-kit", which allows the clinician to select a tool to suit the individual patient and the clinical setting.

An abnormal result on cognitive screening requires further evaluation. Likewise, cognitive concerns in the context of a normal cognitive assessment tool result also warrant further assessment. To contextualize the abnormal cognitive screen, the clinician needs to obtain a broad understanding of a person's level of function and the presence of longitudinal change in cognition [1,6]. A detailed history should be obtained from the patient ideally alongside an informant (usually the patient's family or caregiver), with a focus on the patient's ability to manage personal activities of daily living (e.g., dressing, bathing, and toileting) and instrumental activities of daily living (e.g., household tasks, medication

management, shopping, finance management, and driving) [4]. In addition, validated informant tools can assist in obtaining relevant information [102,105–108]. Driving ability, financial, as well as medication self-management are considered tasks requiring higher cognitive abilities and may be some of the first functional areas where changes are observed in early dementia [109,110]. An assessment of this nature requires time and sensitive inquiry. Moreover, it may be most appropriate for the cancer care clinician to refer the person to a clinician with expertise in geriatric or dementia care.

The assessment can be complemented by an occupational therapist review to assess function-based cognition [4]. A physical examination followed by further pathological and radiological investigations are standard practice for cognitive work-up, to exclude alternate (and possibly reversible) causes of cognitive impairment. A work-up usually incorporates baseline blood tests (full blood count, renal function, electrolytes, liver function tests, thyroid function tests, vitamin B12 levels, calcium level, syphilis serology, HbA1C), urine microscopy culture, and may include cerebral radiological testing [3,111]. Similarly, although outside the scope of this review, the impact of mood disorders on cognitive performance should never be overlooked [112].

Depending on the available services, referral for more comprehensive neurocognitive testing either in a designated memory care clinic or referral to a Geriatrician for a Comprehensive Geriatric Assessment (CGA), whereby other geriatric domains can be reviewed in detail, is recommended [1,113]. Geriatrician input is not only for diagnostic purposes, but also to develop and implement an individualized plan to support the patient and their caregivers to optimize their cancer treatment, based upon the particular areas of deficits revealed [113]. The downstream implementation and integration of GA-guided interventions will differ across organizations. However, the timely communication of the CGA back to the primary treating oncology team to enable the incorporation of this information into the oncological plan is of the utmost importance [14,15].

A limitation of this review pertains to the narrow focus on pre-existing cognitive impairment at cancer diagnosis. It is important to demarcate this from other cognitive conditions associated with cancer, which were outside the scope of this review. Literature on the entity of CRCD is expanding, referring to the deterioration in cognitive domains (in particular memory, attention, concentration, and executive function) during and following the treatment for cancer [4,6]. Again, the screening tools discussed in this review may not be validated for detecting the subtle changes often seen in this pathophysiologically different entity.

There is a high prevalence of cognitive impairment in older cancer survivors, thus while we have considered screening for cognitive impairment at an initial assessment, a longitudinal review is also important [114]. This is not only a consideration in patients who received chemotherapy, but there is growing evidence that hormonal therapies and immunotherapies may also impact on cognition [114]. Similarly, while delirium is a separate entity of acute cognitive decline, it is important to be aware that those with baseline cognitive impairment are at a far higher risk of developing a delirium episode during their cancer treatment phase [1,8]. Education regarding delirium monitoring for these patients and their caregivers is required [10,115,116].

This study was a rapid review rather than a systematic review. We intended for our search to be broad and incorporate various international guidelines to make our study more generally applicable, but we acknowledge that our inclusion criteria required the availability of an English version of the guideline. Moreover, we note that our resulting guidelines originated in countries of similar global economic rankings and did not incorporate developing nations. Similarly, our practicalities and suggestions are more representative of the geriatric oncology practices described in the current geriatric oncology research (and within the geographical context of the Australian authors), which tend to focus on developed countries without incorporating the unique challenges of developing nations.

While we strongly advocate for routine cognitive screening, future research is needed to evaluate the optimal methods for routine cognitive screening and the associated inter-

ventions to support older adults with cancer and their caregivers. Moreover, it would be interesting to explore the varying models of care across geriatric oncology services and how cognitive assessment tools influence the service access and multidisciplinary team involvement in older adults with cancer with pre-existing cognitive impairment.

## 5. Conclusions

Cognitive impairment significantly impacts older cancer patients and their caregivers. International guidelines and position statements for older adults with cancer recommend the routine use of cognitive assessment tools, including the BOMC, CDT, Mini-Cog, MMSE, MoCA, and SPMSQ, to detect cognitive problems. Tool selection is influenced by the clinical environment and patient factors. Clinicians should be familiar with utilizing various tools and their limitations. Every effort should be made to ensure older adults with cancer receive the appropriate cognitive work-up when screening is abnormal.

**Author Contributions:** Conceptualization, H.L., W.K.S., M.A., G.T. and J.L.P.; methodology, H.L., W.K.S., M.A., G.T. and J.L.P.; validation, H.L., K.-Y.L., K.F. and E.L.O.; investigation, H.L., K.-Y.L., K.F. and E.L.O.; writing—original draft preparation, G.T. and H.L.; writing—review and editing, G.T., H.L., W.K.S., M.A. and J.L.P.; visualization, H.L., K.-Y.L., W.K.S. and G.T.; supervision, H.L. and W.K.S. All authors have read and agreed to the published version of the manuscript.

**Funding:** This research received no external funding.

**Acknowledgments:** Technical support, Joshua Dishon.

**Conflicts of Interest:** The authors declare no conflict of interest.

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
