# Peer review of "Cognitive Assessment Tools Recommended in Geriatric Oncology Guidelines: A Rapid Review"

_curroncol, doi:10.3390/curroncol28050339_

Round 1

Reviewer 1 Report

This paper compares a number of key cognitive tests that can be used in an oncology setting. The approach to identify tools was appropriate and the description of the tools was thorough and highlighted the advantages and challenges of each. 

I have no major concerns about this paper.

I do have a few minor suggestions:

Line 72: reducing the chances of misinterpretation of assessment results

Line 194: to improve detection of

Line 208: cognitive domains, but not executive function. 

Line 220: The MoCA was evaluated using patients with dementia of Alzheimer’s type – I would change the wording slightly: It has shown high sensitivity and reasonable specificity in a population of patients with dementia of Alzheimer’s type.

Line 242: such as naming a power…, completing a will and participating in advance….

Line 246: as well as making the logistics

Lines 275 to 279. I would change the order so that the positive statement about MMSE and MoCA having broader coverage goes first. Then state the other two issues about missing domains next.

Line 314: typically driving is included as one of the first IADL domains to be affected

Line 315: and may be some of the first

Line 319: expertise in geriatric or dementia care.

Line 330: we don’t necessarily want each person with cognitive impairment to have a CT – maybe use wording like “may include radiologic testing….”

Line 335: Geriatric Depression Scale

Line 338: “memory care clinic” sounds more positive

Line 364: I believe you meant “with baseline cognitive impairment”

Line 386: recommend

Lines 666 and 668: I would reference the original paper first 

Reviewer 2 Report

Thank you for inviting me to review this manuscript which is a practical review of frequently recommended cognitive assessment tools in oncology guidelines. This type of review is interesting because all too often practitioners get little benefit or use other types of review – such as a psychometric appraisal. I enjoyed reading this manuscript and it is well written. However, I do have some recommendations to strength it further.

From my reading of the flow chart, 17 studies were assessed for eligibility from databases – and 13 were excluded = 4 included. In addition another 18 studies were assessed for eligibility from other sources, and 13 were excluded = leaving a further 5 included. So in total it looks like 9 studies were included – although the flow chart states 8 were included. Is there a discrepancy here, or have I misunderstood? Another point is that the authors have extracted the tools from guidelines – not studies as is suggested in the flow chart – and this is potentially misleading.

Table 5 is almost impossible to read, but this is a minor formatting issue.

My main comment concerns the discussion section. I found much of the discussion section read like what I would expect to find in an introduction/background section (particularly the first two paragraphs). Perhaps the authors should consider amending their manuscript along these lines.

The rest of the discussion was interesting, but I was surprised that the findings were not discussed as much as I thought they should be – particularly around recommendations for which tools are the most desirable. I appreciate there is some writing dedicated to this (and what other factors might determine choice of tool), but the paragraphs on abnormal results and informant tools seem a little superfluous for the context of this review. I read the discussion, and was really none the wiser at what the authors key points were based on the findings. I think the discussion section needs re-crafting with that central issue in mind. The discussion section should really spell out the how the review adds to the literature, and that is lacking with its current focus that is far too wide.

Round 2

Reviewer 2 Report

Thank you for addressing all of my comments, and flagging where you did not agree. The Introduction sections and Discussion are now much more coherent, and overall the paper tells a better story and flows in a much better way.